# Long-read whole-genome analysis of human single cells

Joanna Hård[1,2,3] ✉, Jeff E. Mold[1], Jesper Eisfeldt[4,5], Christian Tellgren-Roth[6], Susana Häggqvist[6], Ignas Bunikis[6], Orlando Contreras-Lopez[7], Chen-Shan Chin[8], Jessica Nordlund[9], Carl-Johan Rubin[10], Lars Feuk[6], Jakob Michaëlsson[11] & Adam Ameur[6] ✉

Long-read sequencing has dramatically increased our understanding of human genome variation. Here, we demonstrate that long-read technology can give new insights into the genomic architecture of individual cells. Clonally expanded CD8+ T-cells from a human donor were subjected to droplet-based multiple displacement amplification (dMDA) to generate long molecules with reduced bias. PacBio sequencing generated up to 40% genome coverage per single-cell, enabling detection of single nucleotide variants (SNVs), structural variants (SVs), and tandem repeats, also in regions inaccessible by short reads. 28 somatic SNVs were detected, including one case of mitochondrial heteroplasmy. 5473 high-confidence SVs/cell were discovered, a sixteen-fold increase compared to Illumina-based results from clonally related cells. Single-cell de novo assembly generated a genome size of up to 598 Mb and 1762 (12.8%) complete gene models. In summary, our work shows the promise of long-read sequencing toward characterization of the full spectrum of genetic variation in single cells.

During the last few years, long-read sequencing technologies have made remarkable progress in terms of throughput and data quality. Due to their capability to read through repetitive and high GC-content regions, these technologies are essential for the ambitious plans to generate reference genomes for virtually all of Earth's eukaryotic biodiversity[1,2], as well as complete telomere-to-telomere maps of the human genome[3]. A further advantage of long-read sequencing is that it facilitates genotyping of complex structural variation (SVs) and repeat elements, which can be difficult or impossible to identify with other genomic sequencing approaches[4–6]. Although clinical long-read sequencing is still in its infancy[7,8], several studies have already demonstrated the potential to discover novel disease-causing human

genetic variation[9]. Long sequencing reads also enables the detection of clinically relevant genetic variation in 'dark DNA', representing regions of the human genome that cannot be analyzed with standard short-read technologies[10].

Long-read sequencing holds many promises, but one research area that remains unexplored is single-cell genomics. Human single-cell whole-genome sequencing (WGS) emerged about a decade ago[11–15], and has become an active field of research with the potential to answer fundamental questions in several areas of cell biology, such as somatic genetic variation[16], tumor evolution[11], de novo mutation rates[14], meiotic recombination of germ cells[14,17], or neurogenetics[18–20]. Until now, single-cell WGS projects have focused on characterizing genetic

[1]Department of Cell and Molecular Biology, Karolinska Institutet, Stockholm, Sweden. [2]Department of Biosystems Science and Engineering, ETH Zurich, Basel, Switzerland. [3]ETH AI Center, ETH Zurich, Zurich, Switzerland. [4]Department of Molecular Medicine and Surgery, Karolinska Institutet, Stockholm, Sweden. [5]Department of Clinical Genetics, Karolinska University Hospital, Stockholm, Sweden. [6]Science for Life Laboratory, Department of Immunology, Genetics and Pathology, Uppsala University, Uppsala, Sweden. [7]Science for Life Laboratory, Royal Institute of Technology (KTH), Stockholm, Sweden. [8]GeneDX LLC, Stamford, CT 06902, USA. [9]Science for Life Laboratory, Department of Medical Sciences, Uppsala University, Uppsala, Sweden. [10]Department of Medical Biochemistry and Microbiology, Uppsala University, Uppsala, Sweden. [11]Center for Infectious Medicine, Department of Medicine, Karolinska Institutet, Stockholm, Sweden. ✉e-mail: joanna.hard@bsse.ethz.ch; adam.ameur@igp.uu.se

variation detectable from short-read Illumina sequencing protocols[21–25], including single nucleotide variants (SNVs)[19,21,26–31], SVs[32], large-scale copy number variation[30,33–35] and retrotransposon elements[12,18,36,37]. While multiple methods exist for single-cell short-read WGS, there are no available technologies to support the identification of the full extent of genetic variation in individual cells. One recent report demonstrates the utility of long-read sequencing for analysis of SVs, but with limited performance in the detection of other types of genetic variation[38].

The lack of methods for single-cell WGS with long-read technologies can in part be explained by the throughput of long-read sequencing instruments, which until recently has been relatively modest. Moreover, single-cell genome sequencing is challenging[39] since a diploid cell contains only two DNA molecules at each locus in the genome, and every molecule that is lost during sample preparation, or fails to be sequenced, inevitably leads to allelic dropout and missing data. Furthermore, the long-read sequencing protocols require large amounts, typically several micrograms, of input DNA. This is about a million times more DNA than what is contained within a single human cell, which implies that substantial DNA amplification is required.

Whole-genome amplification (WGA) has a profound detrimental effect on the sequencing results and should be avoided when possible. This is because WGA introduces amplification bias, chimeric molecules, and allelic dropout. Several different amplification protocols have been developed[15,40,41] and it is crucial to choose a method that minimizes artifacts and biases, while at the same time being compatible with the downstream sequencing technology. Multiple displacement amplification (MDA)[41] has the capacity to amplify kilobase-length molecules and could therefore a suitable approach for long-read sequencing. With regards to amplification bias, it has been proposed that a droplet-based MDA (dMDA) reaction, performed on DNA fragments contained within nano- or picoliter droplets, can minimize differences in amplification gain among the fragments[42–44]. Such a droplet-based amplification could also be an efficient approach to remove inter-molecular chimeras since MDA artifacts only can be formed between molecules contained within the same droplet.

Single-cell DNA fragments amplified by MDA are well-suited for PacBio high-fidelity (HiFi) sequencing[45], a protocol that works best for molecules of up to 20 kb in length. HiFi reads have high accuracy (>QV20) and allow not only for the identification of complex genetic variation such as SVs and repeat elements, but also SNVs at a level that matches the ability of short-read sequencing[45]. PacBio HiFi sequencing has also proven to be an excellent method for high-quality genome assembly[46–49], thereby raising the prospect of long-read de novo assembly of genomic DNA from individual cells[50]. The potential applications are not limited to human cells. Long-read WGS could also potentially generate improved genome information for other sample sources, such as single-cellular organisms that are difficult to culture.

In this study, we present a new method for long-read whole-genome analysis of human single cells. Our workflow utilizes an automated dMDA technique for single-cell WGA coupled with PacBio HiFi WGS. The method was evaluated on clonally expanded CD8+ T-cells from a human donor. For comparison, other cells from the same T-cell clones were sequenced with short-read Illumina WGS. Our results show that long-read sequencing gives improved performance for the detection of genetic variation in single cells, in particular for haplotype phasing, complex structural variation, and tandem repeats (TR). Moreover, we show that it is possible to reconstruct parts of a human T-cell genome de novo.

## Results

### A single-cell WGS workflow compatible with short- and long-read sequencing

We first aimed to develop a DNA amplification method that preserves molecule lengths and reduces amplification bias (Fig. 1a). Briefly, one single cell is isolated by fluorescence-activated cell sorting (FACS) and placed into a well containing lysis buffer, so that the DNA fragments are released. The DNA molecules are then encapsulated in ~50,000 droplets, after which a dMDA reaction takes place within each droplet. The droplets have a diameter of <100 μm and are generated using the Xdrop system[51] (Fig. 1b). Only one or a few DNA fragments will be located in each droplet, and since the amplification takes place in a small volume containing limited reagents this prevents molecules from being heavily over-amplified. Moreover, the risk of forming inter-molecular chimeras during the dMDA reaction is greatly reduced by our method. In droplets harboring a single DNA fragment, the risk is completely eliminated. Once the dMDA reaction is complete, the amplified DNA can be used for the preparation of short- or long-read sequencing libraries. For our experiments, two individual CD8+ T cells (A and B) from the same human donor were clonally expanded in vitro, and the resulting cell collections were used as starting material for WGA and sequencing (Fig. 1c). In addition, bulk DNA isolated from peripheral blood mononuclear cells (PBMC) obtained from the same individual was analyzed, at over 30× WGS coverage both with short Illumina reads and long PacBio HiFi reads. Since the bulk datasets represent millions of cells from the same individual, they should contain all germline variation detectable by the respective technology and can therefore be used as germline 'truth sets' for both clonal expansions. A limitation with using PBMCs as ground truth is that signals from somatic variants present in founder cells of the individual clonal expansions will be concealed within the bulk data. While such somatic variants cannot be validated by the bulk sample, they will not have a noticeable impact on the overall performance statistics given the low number of somatic variants as compared to germline variation.

### dMDA increases coverage uniformity

Sixteen single-cell DNA samples from the two T-cell clones A and B were analyzed using Illumina WGS. Eight of the samples were amplified using dMDA, while the remaining eight samples were subjected to standard MDA (MDA). The sequencing resulted in 100–200 million read pairs per sample (Supplementary Table S1), and these were aligned to the GRCh38 human reference build. Quality control was run on all single-cell samples and the results are available (supplementary Data 1). To facilitate direct comparisons between the samples, all Illumina datasets were randomly subsampled to contain ~100 million read pairs (supplementary Table S2). As expected, the eight dMDA samples displayed a more uniform coverage across the genome as compared to the eight MDA samples (Fig. 2a–c). Furthermore, our results revealed that the uneven coverage in the MDA samples originated from a limited number of fragments that were amplified to extreme coverage (Fig. 2d). For the MDA samples, on average 68.9% of the reads aligned to regions with ≥200× coverage, while the corresponding percentage for dMDA was only 16.0%. In these downsampled datasets, 33.8% of bases were covered by at least one read in dMDA as compared to 23.4% for MDA (Fig. 2e. Coverage variation across the genome was further quantified for the complete single-cell datasets, and the standard deviation was found to be 2.46 times higher for MDA as compared to dMDA samples (supplementary Table S3 and supplementary Fig S1). Based on these results, we conclude that dMDA gives increased sequencing coverage uniformity as compared to regular MDA, thereby corroborating previous evaluations of droplet-based MDA methods[42,43].

### Long-read whole-genome sequencing of individual T-cells

We used five dMDA single-cell samples, two from T-cell clone A and three from T-cell clone B, for PacBio long-read sequencing (supplementary Table S4). The dMDA reactions generated 1–4 μg of amplified DNA and the fragment size distributions displayed peaks ~9 kb (supplementary Fig S2). We subsequently prepared libraries from the amplified DNA using a bead-based size selection (see Methods) and sequenced each library on an individual 8 M SMRT cell, generating up

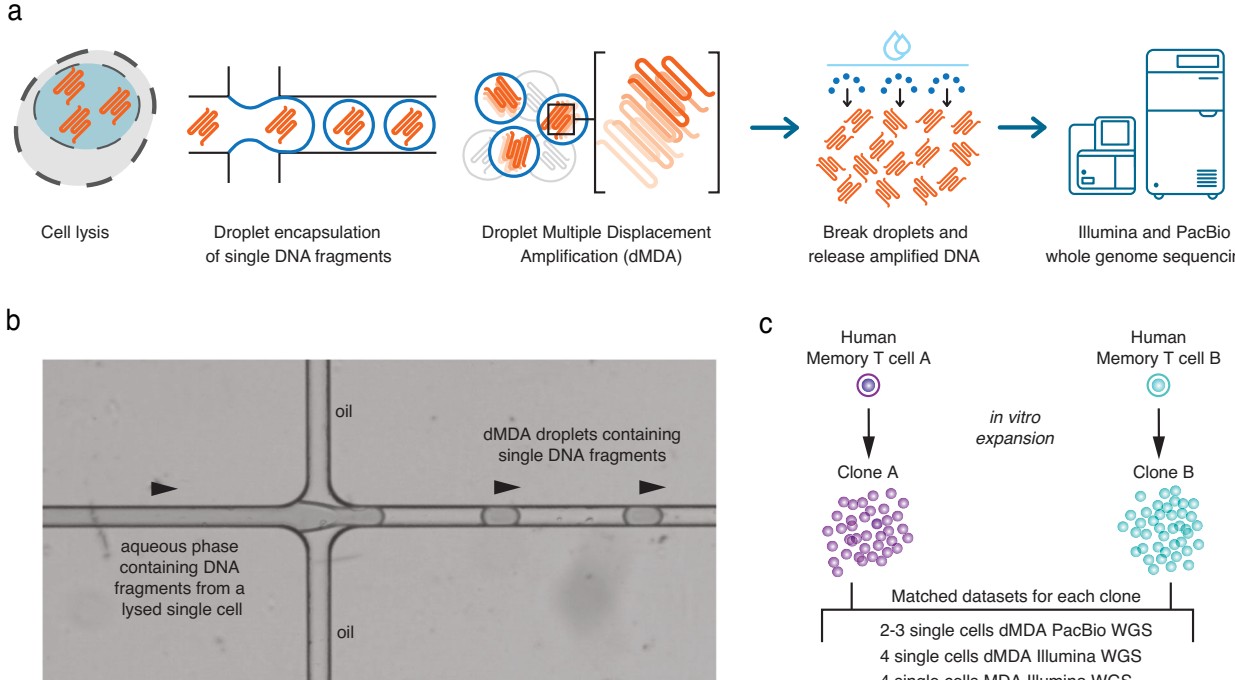

**Fig. 1 | Overview of the single-cell DNA amplification and sequencing experiment. a** An individual cell is isolated by fluorescence-activated cell sorting (FACS) and placed into a well-containing lysis buffer. DNA molecules from the lysed single cell are then encapsulated in picoliter droplets using the Xdrop microfluidic system, after which dMDA whole-genome amplification takes place inside each droplet. After amplification, the droplets are broken and DNA is released, followed by library preparation and whole-genome sequencing using short- (Illumina) and long-read (PacBio) technologies. **b** Image showing how droplets are formed in the Xdrop microfluidic system. An aqueous phase containing lysed DNA and dMDA reagents encounters an oil layer, resulting in <100 μm diameter droplets where single DNA fragments are captured. The Xdrop system has the capacity to produce around 50,000 droplets in 45 s. **c** Two human memory T cells (cells A and B) from the same individual were used as starting points for the experiments. Collections of daughter cells were obtained by in vitro expansion, and individual cells from clones A and B were analyzed using Illumina and PacBio whole-genome sequencing.

to 2.75 million reads and 20 Gb HiFi data (>QV20) for a single cell sample (supplementary Table S5). About 99% of reads could be aligned to GRCh38 and the average alignment concordance was 98.91% or higher. On average, 2.7 separate alignments were produced for each HiFi read, indicating the presence of chimeric artifacts from the dMDA. These chimeric reads are mainly intramolecular, i.e., formed within encapsulated DNA molecules during the dMDA reaction, and they can be split into non-chimeric parts using bioinformatics tools. The aligned read length is a good indicator of the non-chimeric part of a read since it corresponds to the longest subsequence that can be continuously matched to the GRCh38 reference. The N50 aligned read length was 2.8–3.6 kb for the single cells, and the maximum read alignment was over 50 kb.

### Mitochondrial heteroplasmy in T-cell clone B

mtDNA represents a substantial source of somatic genetic variation[52]. However, amplification bias may result in limited read coverage over mtDNA and prevention to detect mitochondrial variation. Based on our single-cell Illumina data we found that dMDA outperforms MDA in terms of mtDNA coverage (Supplementary Fig S3). In the long-read data, we found that 17% of the mtDNA reads covered at least 25% of the complete mitochondrial genome, thereby opening up the possibility to phase large parts of the individual mtDNA molecules present in a single cell. In our long-read datasets, we observed one location (chrM:16,218) where a C > T nucleotide substitution occurred in 41%–67% for the three single cells from T-cell clone B, while being completely absent from the reads for T-cell clone A, as well as from the bulk DNA sample (supplementary Fig S4). By further analyzing the Illumina data for the two single-cell clones, we validated that the nucleotide substitution was present in T-cell clone B, but not in T-cell

clone A, consistent with mitochondrial heteroplasmy in T-cell clone B (supplementary Fig S5).

### Detection of SNVs in single-cell long-read data

15.7 Gb data was obtained for the PacBio single cells on average, as compared to 48.7 Gb for Illumina (Fig. 3a), and this data was used for single-cell SNV calling. Between 0.89 and 1.32 million SNVs were detected in the single-cell PacBio data by the software DeepVariant[53] (supplementary Tables S6, S7). An average of 0.88 million SNVs per single cell were found to be overlapping with SNVs called from the bulk PacBio HiFi data. Since DeepVariant has been shown to achieve genome-wide SNV precision and recall rates of at least 99.91% on PacBio HiFi data[45], we consider these overlapping SNVs to be true (Fig. 3b). For comparison, a similar analysis using Illumina single-cell dMDA and bulk data resulted in an average of 1.06 million true SNVs per cell. This implies that a similar number of germline SNVs were detected using PacBio as compared to Illumina, despite that PacBio single-cell sequencing only generated 32% of the Illumina data amount on average. We further estimated the precision and sensitivity of the SNV calls (Figs. 3c, d). PacBio single-cell SNV calling resulted in higher precision than what was obtained for Illumina dMDA (0.86 vs 0.74). In contrast, the sensitivity of PacBio SNVs was somewhat lower than for Illumina dMDA (0.17 vs 0.24). The overall low sensitivity can be explained by the high level of allelic dropout in the single cells, in particular for the PacBio samples where less data is available. A total of 284k high-confidence PacBio SNVs/cell escaped detection in the Illumina bulk sample (supplementary Table S8). Of these variants, 6336 were located in previously reported "dark" genic regions of relevance for human health[10]. One such region comprises introns and exons of *NBPF8* (Fig. 3e). Another example is *CDC73*, where a repeat resolved in

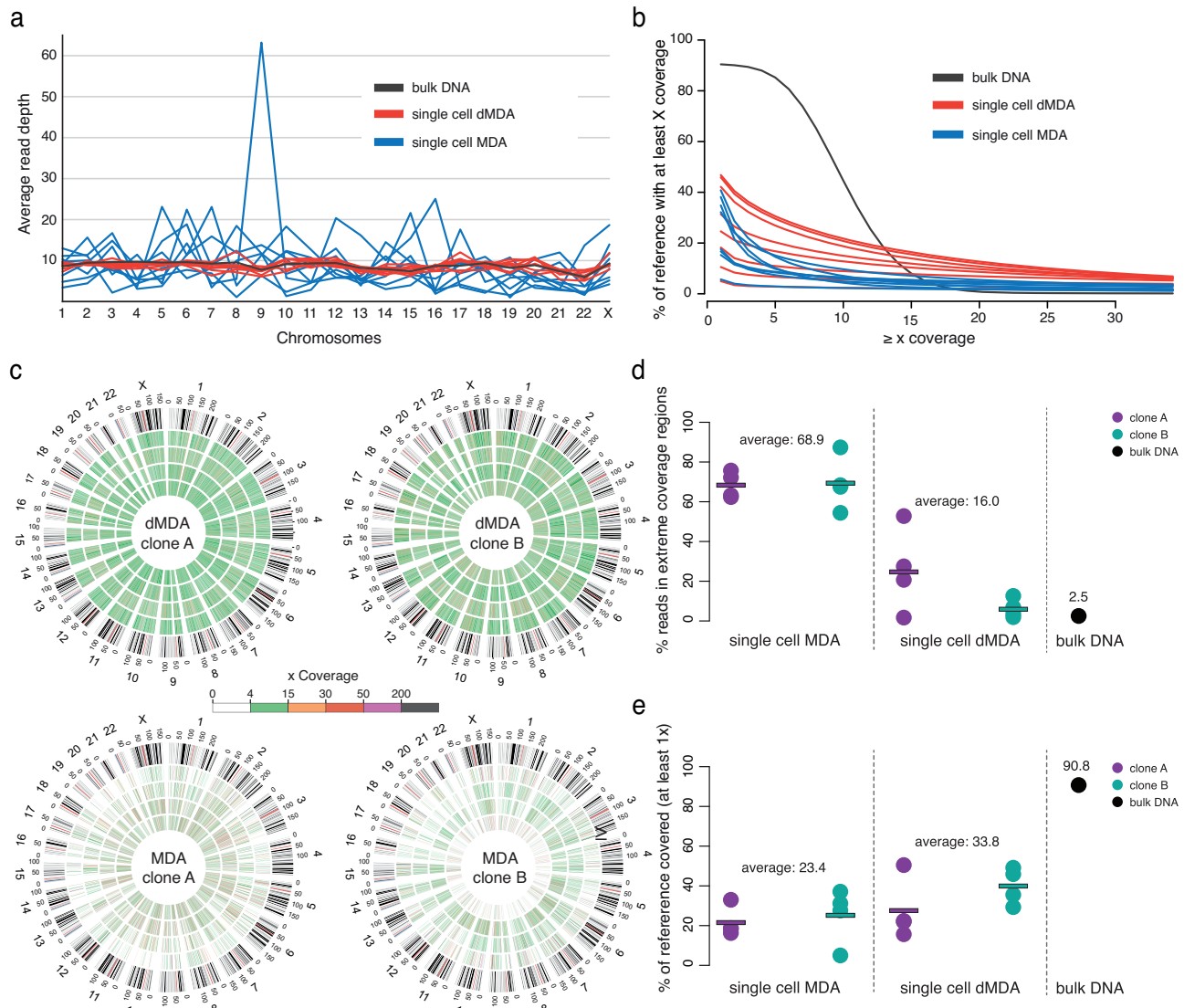

**Fig. 2 | Comparison of MDA and dMDA for whole-genome amplification.** These results are based on Illumina MDA, dMDA, and bulk sequencing where the datasets have been randomly downsampled to contain the same number of reads. **a** The figure displays the average sequencing depth across the human chromosomes. The dMDA single-cell samples display good uniformity of coverage, whereas the MDA data show high spikes due to amplification bias. **b** Plot showing the percentage of bases in the reference genome (y axis) having a minimal coverage (x axis). On average the dMDA samples have more bases covered at a range 10–30×, as compared to the single-cell samples subjected to regular MDA. **c** Circle plots showing sequencing coverage in 500 kb bins for all of the Illumina single-cell samples, color-coded from 0× coverage (white) to over 200× coverage (black). Four replicate samples are included in each of the circle plots, and the chromosomal coordinates are displayed in the outermost circle. The dMDA samples at the top row display more even coverage than the MDA samples below, with more of the bins having average coverage in 4–15× coverage range (green). **d** Dot plot showing the percentage of reads aligning to regions of extreme (≥200×) coverage. Each dot corresponds to an individual sample. The rectangles indicate the average values for different sample/clone combinations (n = 4 cells per group). 68.9 and 16.0 are the average values for all MDA and dMDA samples (n = 8 cells per group). **e** Dot plot showing the percentage of reference bases that are covered by at least one read. The rectangles indicate the average values for different sample/clone combinations (n = 4 cells per group). 23.4 and 33.8 are the average values for all MDA and dMDA samples (n = 8 cells per group). Source data are provided as a Source Data file.

the PacBio single-cell data was represented as an alignment gap in the Illumina bulk data (Fig. 3f). In addition to germline SNVs, 27 somatic SNVs were identified in the PacBio data (supplementary Tables S9–S10). The somatic SNVs were present in at least two single cells from one of the clones, while being absent from the other clone and from the bulk sample (see Methods). Two examples of somatic SNVs are shown in Fig. 3g, h. It is important to note that the PacBio data provides phasing information over several kilobases. This makes it possible to assign variants detected in the single cells to one of the haplotypes, which highlights one of the advantages of long-read sequencing for the identification and analysis of somatic variation at the single cell level[26,27].

## Long-read single-cell WGS improves the detection of structural variants

Structural variants (SVs) were called from the PacBio data using Sniffles2[54]. This resulted in 3493 deletions, 4636 insertions, 903 duplications, and 19373 inversions per single cell, on average (Fig. 4a, supplementary Tables S11–S15). The SVs were merged across samples so that the true SVs, i.e., overlapping with the bulk sample, as well as the SVs unique to the single cells could be identified. Over 80,000 of the SVs in the PacBio single cells were not detected in the bulk sample, and the vast majority of these are likely to originate from chimeric dMDA molecules. 81.5% of these false SVs were inversions, 9.0% insertions, 5.0% duplications, and 4.6% deletions, and their lengths

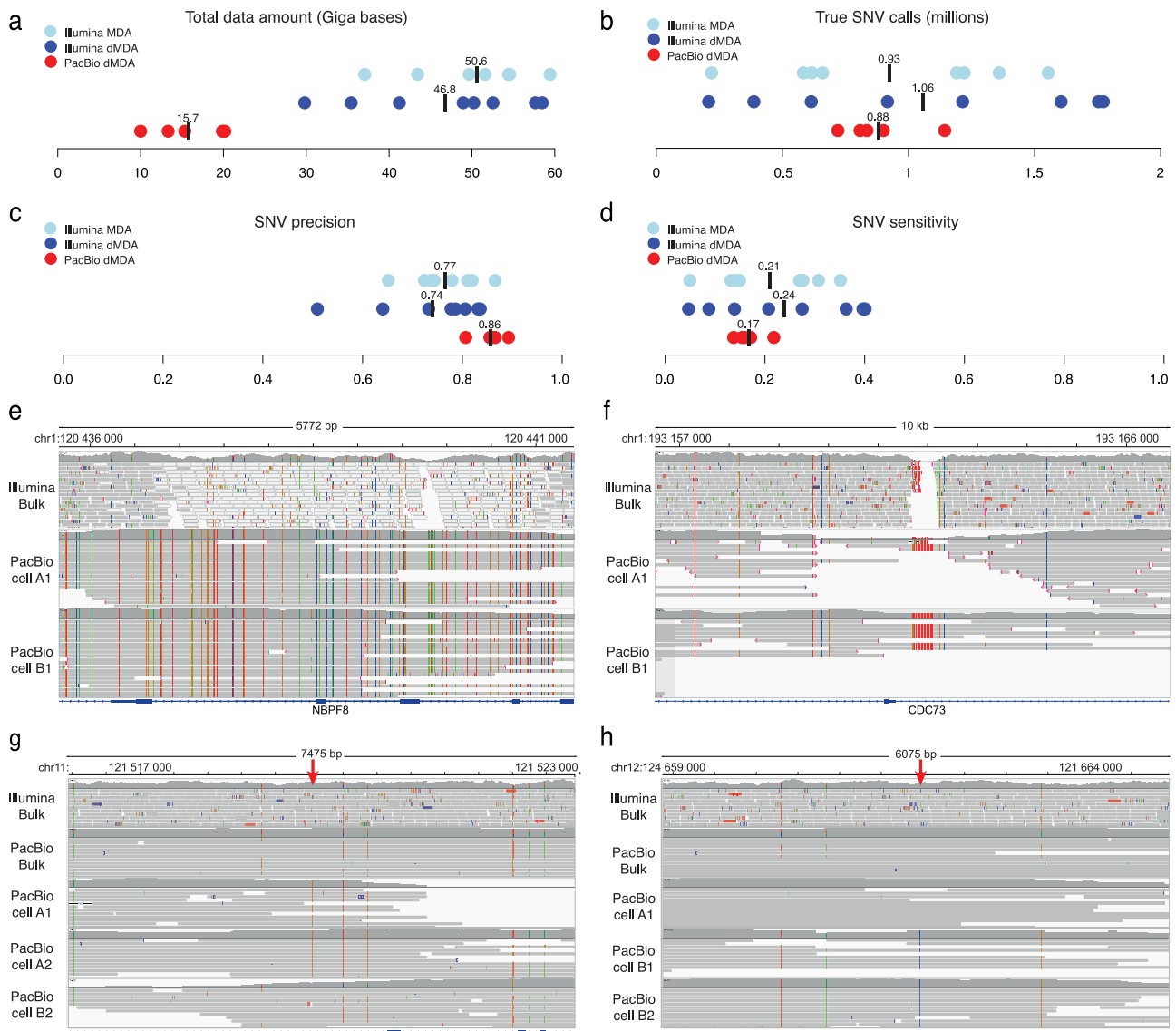

**Fig. 3 | Analysis of SNVs in short- and long-read single-cell data. a** Total data amount for the Illumina and PacBio single-cell samples. Average values are represented by black vertical lines. **b** Number of true positive SNV calls in the Illumina and PacBio single cells. The true SNVs are defined as those found to be present also in the corresponding bulk sample. **c** Precision of SNV calls in the single-cell samples. **d** Sensitivity of SNV calls in the single-cell samples. **e** Example of a "dark" genic region (*NBPF8*) where Illumina data fails to align uniquely, while SNVs can be identified and phased in the PacBio single-cell data. **f** Another example of a "dark" genic region (*CDC73*), where PacBio reads from the two single cells span across a

repetitive region that lacks coverage in the Illumina bulk sequencing data. **g** A somatic SNV in an intron of *SORL1*. The position of the somatic SNV is indicated by the red arrow at the top. The PacBio A1 and A2 single-cell samples contain a C > G variant at this position that is linked to several nearby SNVs in the region. In the bulk and PacBio B2 samples, the G is absent from the haplotype. This indicates that the C > G is a somatic variant only present in the T-cells from clone A. **h** A somatic SNV in an intergenic region on chromosome 12. This G > C variant is only present in T cells from clone B. Source data are provided as a Source Data file.

were usually <5 kb (supplementary Fig S6). An average of 5473 true SVs were detected per single cell, of which 2510 were deletions and 2957 insertions, but only a handful of true duplications or inversions were detected (Fig. 4b). For comparison, we performed SV analyses also in the Illumina samples (supplementary Table S16). In the Illumina dMDA samples, we detected 327 true SVs on average, which is <1/16 of the number of true SVs found in PacBio single cells. The precision for PacBio deletions and insertions was 0.73 and 0.66, respectively, while their sensitivity was slightly above 0.20 (Fig. 4c, d). Duplication and inversions had a precision near zero since virtually all such events originate from chimeric molecules. The true PacBio SVs consisted mainly of insertions and deletions of up to 1 kb length, with clear peaks ~300 bp representing ALU repeat elements (Fig. 4e). However, we also detected a peak for deletions around 6 kb corresponding to LINE

elements (Fig. 4f). Several of the SVs detected in the single-cell PacBio data are difficult to identify in Illumina bulk data, e.g., insertion of 710 bp and a deletion of 4891 bp (Fig. 5). The PacBio data not only allows us to determine the exact breakpoints for these SVs in single cells, it also enables the reconstruction of the haplotypes through phasing of nearby heterozygous SNVs. We further searched for somatic SVs in the long-read data but found no events that differed between the T-cell clones.

### Analysis of TRs in single cells
We hypothesized that the PacBio reads would allow us to study TRs in single cells. To investigate this, we performed TR calling using Tandem Genotypes[55]. We first extracted all TRs that were called either homo- or heterozygous in the PacBio bulk data and found

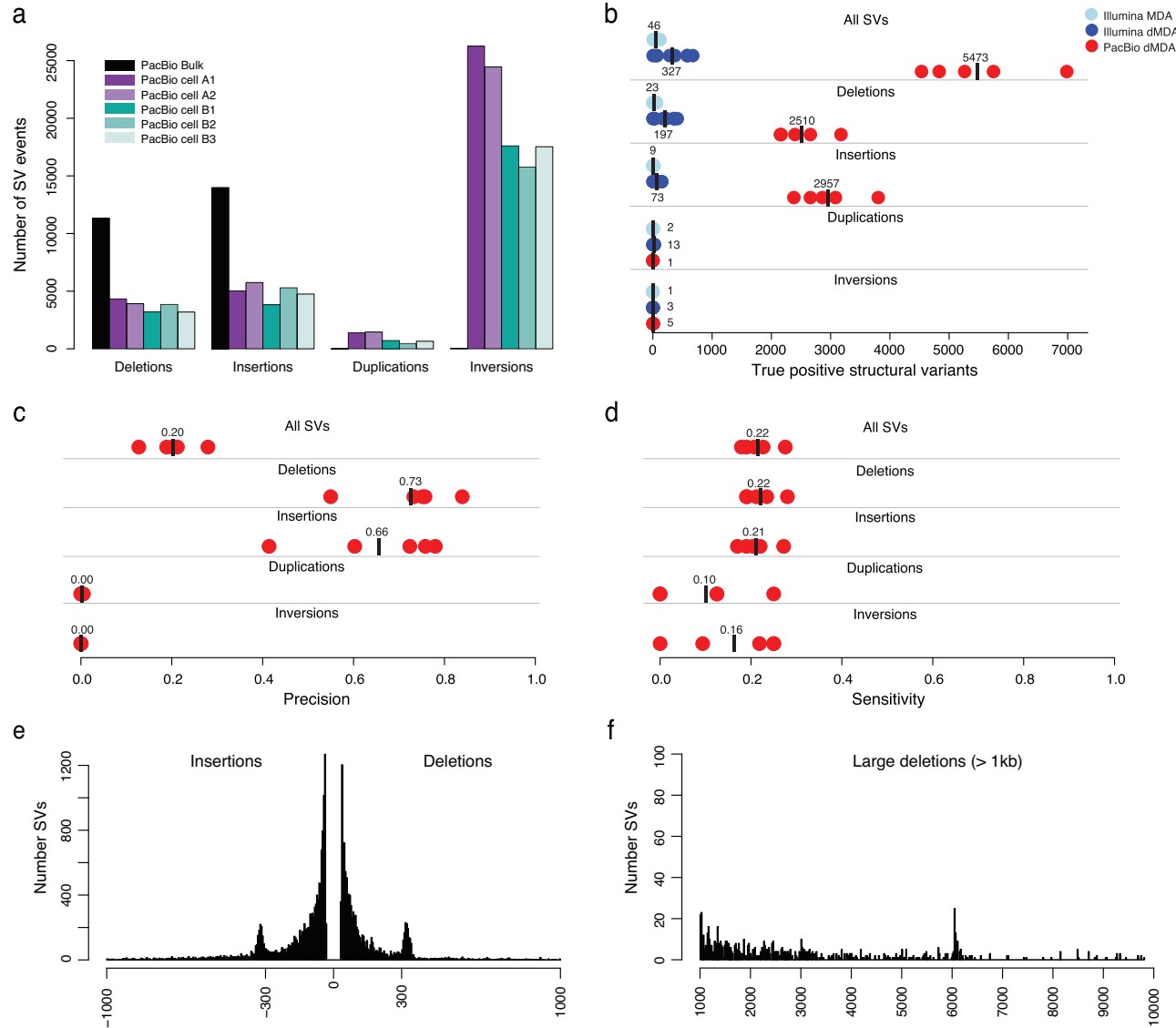

**Fig. 4 | Analysis of SVs in long-read single-cell data. a** Number of deletions, insertions, duplications, and inversions were detected by Sniffles2 in all five PacBio single-cell datasets from the T-cell clones A and B. For comparison, the black bars represent SVs detected in the PacBio HiFi bulk sample. Many more duplications and inversions are detected in the single-cell data than in the bulk sample, as a result of chimeras introduced by dMDA. **b** Number of true positive SV calls in the Illumina and PacBio single cells. The true positive events are defined as those overlapping with an SV from the corresponding bulk sample. Average values for each SV class are represented by black vertical lines. **c** Precision of single-cell SV calls for the five PacBio single cells. Duplications and inversions have a precision of zero since virtually none of these events are detected in the PacBio bulk DNA sample. **d** Sensitivity of single-cell SV calls for the five PacBio single cells. **e** Length distribution of all true positive inversion and deletion events of up to 1 kb detected in the PacBio single cells. **f** Length distribution of all true positive deletion events of length 1–10 kb detected in the PacBio single cells. Source data are provided as a Source Data file.

15,098 such repeat elements. In the single cells, an average of 4770 TR alleles could be correctly genotyped with a repeat size in agreement with the bulk sample (supplementary Table S17). The size distribution of these repeat alleles across all single cells is shown in Fig. 6a. The longest repeat was 662 bp longer than the reference genome and mainly consisted of dinucleotide AT sequences, a region that was difficult to resolve in the short read data (Fig. 6b). We found no clear evidence of clonal somatic variation of repeat elements in our single-cell long-read data. However, it should be noted that this analysis does not give information about all the repeats in the single cells, in particular those of length >500 bp which were largely missing from our results due to unsuccessful genotyping. A common reason for failed TR genotyping was the presence of >2 repeat lengths in the same sample, making it difficult to determine the exact TR sizes.

### De novo assembly of single-cell genomes from long-read data

PacBio HiFi reads are ideal for generating high-quality assemblies of human genomes[45–49], and we were interested to investigate to what extent regions of the single-cell genomes could be reconstructed de novo. We therefore selected the two single cells with the highest coverage in clone A and B and performed an individual de novo assembly for each of these single cells. Since assembly of single-cell PacBio data is challenging due to allelic dropout and chimeric reads, we developed a filtering method to remove chimeric reads from the dataset prior to assembly. Because of the dMDA, chimeras are mainly formed within the same molecule, and by screening each read for inverted or duplicated elements, chimeric reads could be identified and removed ab initio (see Methods). For T-cells A1 and B1, 44.2% and 46.6% of PacBio reads, respectively, passed our filtering criteria. However, this filtering is stringent and while effectively removing

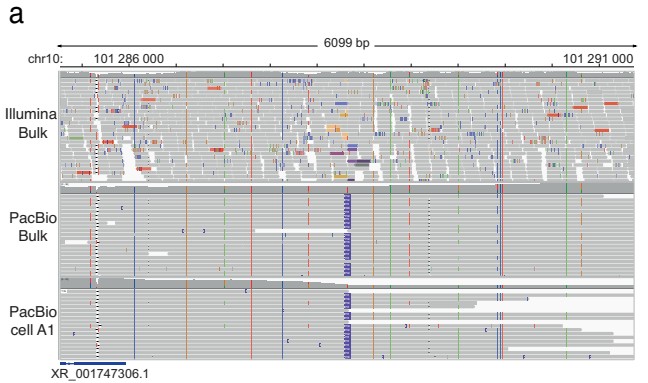

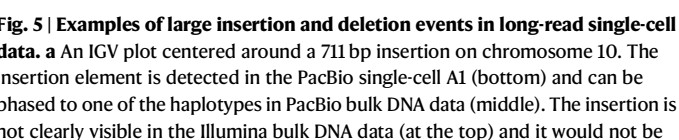

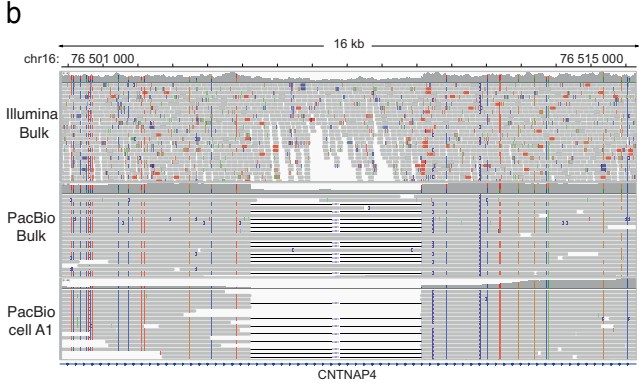

**Fig. 5 | Examples of large insertion and deletion events in long-read single-cell data. a** An IGV plot centered around a 711 bp insertion on chromosome 10. The insertion element is detected in the PacBio single-cell A1 (bottom) and can be phased to one of the haplotypes in PacBio bulk DNA data (middle). The insertion is not clearly visible in the Illumina bulk DNA data (at the top) and it would not be possible to resolve the haplotypes from short-read sequencing alone. **b** A 4891 bp deletion in an intron of *CNTNAP4* was identified both in PacBio single-cell and bulk data. The Illumina data shows a drop in coverage indicative of a heterozygous deletion, but the exact breakpoints and haplotypes are not clearly visible.

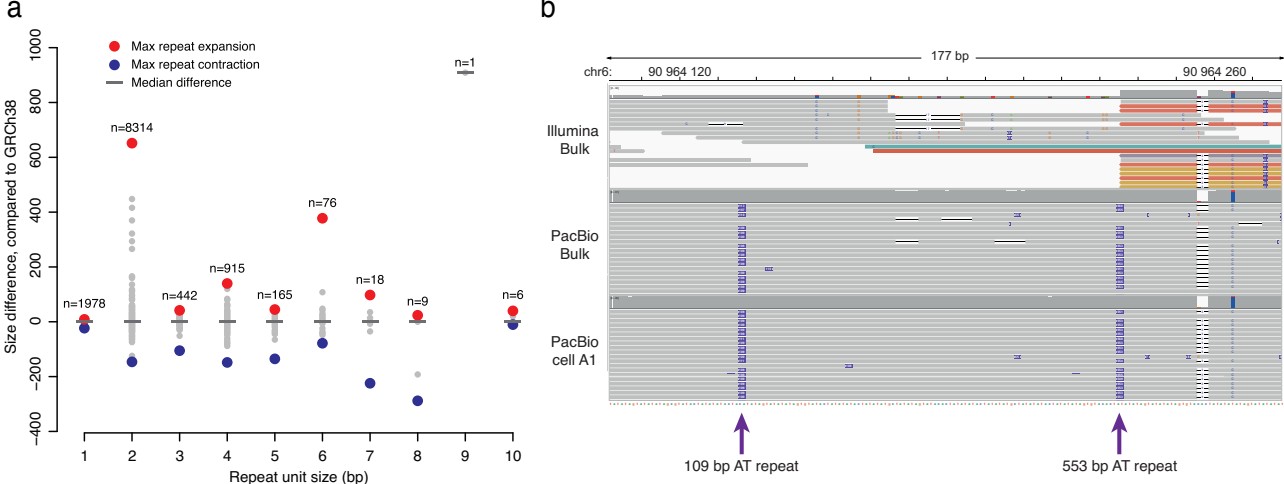

**Fig. 6 | Tandem repeats detected in single-cell long-read data. a** Overview of high-confidence tandem repeats detected across the five single-cell samples. All of these repeats were identified with the same repeat size in the PacBio bulk DNA sample. On the *x* axis is the repeat unit size and on the *y* axis is the length difference of the repeat, when compared to the human GRCh38 reference. The *n* values indicate the total number of analyzed repeat elements of different sizes. A negative value on the *y* axis corresponds a contraction of the repeat and a positive value corresponds to a repeat expansion. The extreme values are marked in red and blue. The median differences (indicated by gray vertical lines) are zero for most of the repeat units, meaning that most of the repeats have the same size as in GRCh38. **b** The IGV plot shows a region on chromosome 6 where the single-cell data an AT-repeat of 662 bp was detected in the PacBio single-cell and bulk data. In the alignment, the 662 bases are divided into two separate repeat insertions with 109 bp and 553 bp, respectively. The region is difficult to analyze by Illumina sequencing.

chimeras, it also removes some correct reads harboring repeat elements. Hifiasm[56] generated primary assemblies of size 598.3 Mb for T-cell A1 and 454.1 Mb for T-cell B1, corresponding to approximately 19% and 15% of the human reference (supplementary Table S18). The contig N50 values were 35 kb (T-cell A1) and 42 kb (T-cell B1), and the largest contig of 578.3 kb was detected in the T-cell B1 assembly. In addition, ~40 Mb of alternative contigs were found in each sample. These alternative contigs correspond to regions where hifiasm reported two distinct haplotypes. We further performed an analysis of BUSCO gene models[57] and could conclude that 12.8% of genes (*n* = 1762) were completely assembled for T-cell A1, and 9.0% of genes (*n* = 1236) for T-cell B1. Complete mitochondrial genomes were obtained and these were identical for T-cells A1 and B1. Detailed quality assessment results for the two single-cell genome assemblies are available in supplementary Table S19.

## Discussion

By combining methods for automated single-cell processing and droplet-based WGA with PacBio HiFi sequencing, we were able to sequence long DNA fragments from single human T-cells. The long sequencing reads resulted in improved analyses of genetic variants as compared to short-read technologies, including in "dark" regions of the human genome, and even enabled de novo assembly of parts of the single-cell genomes. The single cells used as the starting point for this study were obtained through in vitro expansion of CD8+ T-cell clones from a healthy human donor. Studying normal somatic cells isolated from a healthy human donor is more challenging as compared to cells from an immortalized cell line, but with the advantage of being more biologically relevant.

Despite that the yield of the PacBio sequencing was only one-third of the average data amount generated for the single cells on the

Illumina platform, we detected similar numbers of SNVs in PacBio HiFi data and Illumina data. The PacBio SNV precision of 0.86 was slightly higher than what was obtained for Illumina. From the PacBio data, we found 28 somatic SNVs that distinguish the two in vitro expanded T cell clones, including one example of mitochondrial heteroplasmy. This finding demonstrates the utility of our method to detect genetic mosaicism at the single-cell level. Furthermore, the PacBio HiFi data enables the identification of genetic alterations that are difficult to detect in Illumina short-read data. 5473 true positive SVs were found per single cell, with a precision of 0.66 for insertions and 0.73 for deletions. We also detected numerous inversions and duplications in the single cells, but virtually all of these were found to be false positives caused by chimeric molecules introduced during dMDA. The current version of this method is therefore not suitable for studying inversions or duplications. TRs are of particular interest since they have not previously been explored in single cells. On average our method determined the correct size for 4770 TR elements per single cell, several of them of length over 100 bp.

Although somatic variation in SVs and TRs may exist, we found no such evidence in this study. Our results thus indicate that the genetic differences between the two T-cell clones A and B mainly consist of SNVs and that more complex structural variation occurs at a lower frequency. However, more detailed analyses of larger numbers of single cells from the two T-cell clones would be required to give conclusive answers about the full extent of somatic variability in SVs and repeat elements. Moreover, our study is still limited by the non-chimeric read lengths generated by the dMDA, and even longer reads would be required to investigate many of the larger repeats and complex SVs that may occur in these genomes.

A human cell contains about six picograms of DNA, and this is a major challenge for PacBio sequencing which typically requires several micrograms of input material. Thus, substantial amplification of single-cell genomes is required to obtain sufficient amounts of DNA. Several approaches for single-cell WGA exist[38,58–60]. Other methods could potentially give a much more complete representation of the genome[60] than what we obtained in this study, although the amplification results may also differ depending on the cell type. However, the majority of WGA methods are not suited for long-read sequencing due to limited fragment lengths obtained after amplification. MDA represents a promising approach for WGA and long-read sequencing since this method can produce 10–12 kb long fragments. However, MDA is known for amplification bias which, in turn, limits the complexity of sequencing libraries so that only a small fraction of the genome may be recovered. Here, we take advantage of MDA to produce long fragments while reducing the amplification bias by performing MDA reactions in individual droplets. In this study, we used the Xdrop system for dMDA, but the methodology should be possible to adapt also to other droplet generator platforms.

Allelic dropout is always a challenge for single-cell WGS, and our results could be improved by having a higher proportion of DNA fragments encapsulated and amplified in the droplets. Several factors could lead to allelic dropout, including failed amplification of DNA molecules lost in sample preparation, droplet reagent loading, or DNA fragments that are either too short or too long to be efficiently encapsulated. Another important challenge is that WGA introduces chimeras and errors. To some extent, this might be improved by alternative amplification methods or modified experimental conditions. However, further optimizations may not enable the complete removal of amplification errors in the resulting reads. An increased amount of genomic information from single cells may instead be achieved through the development of new bioinformatics tools, specifically designed for single-cell long-read WGS data, which can resolve amplification errors and maximize the utility of the data.

In this project, we opted for PacBio HiFi sequencing since it currently offers the highest per-read accuracy[61]. Although nanopore WGS[62] could be an alternative, it would likely be more challenging to study SNVs and identify chimeric artifacts from nanopore reads because of their higher error rate. However, there is still an open question about which sequencing platform would be best suited for this application in the future. This will depend on factors such as the sequencing yield, quality, and cost per sample, for coming versions of instruments. Due to the rapid developments of long-read technologies, we anticipate that several of these parameters can be radically improved over the coming years.

In conclusion, we demonstrate that long-read genome analysis can be performed not only at a species, population, or individual level but also for single cells. Ultimately, new innovations and technical advances may in the future enable near-complete genome assemblies and full haplotype reconstructions from individual cells.

## Methods
### Single-cell samples
Peripheral blood was collected from a healthy human donor by veni-puncture. The donor was previously vaccinated with the live, attenuated Yellow Fever Virus (YFV) vaccine (YFV-17D) as part of an ongoing study to investigate the dynamics of adaptive immunity to YFV vaccination. Informed consent was obtained from the donor and the study was approved by the Regional Ethical Review Board in Stockholm, Sweden: 2008/1881-31/4, 2013/216-32, and 2104/1890-32. To expand CD8+ T cell clones from single YFV-specific memory CD8+T-cells, mononuclear cells were isolated from peripheral blood by density centrifugation and were first stained with HLA-A2/YFV(LLWNGPMAV)-dextramer FITC (Immudex, Denmark) for 15 min at 4 °C, followed by staining with anti-CD8a-BV570 (clone RPA-T8, Biolegend), anti-CD3-PE/Cy5 (clone UCHT1), anti-CD14-V500 (clone MφP9), anti-CD19-V500 (clone HIB19) (all from BD Biosciences), and LIVE/DEAD™ Fixable Aqua Dead Cell Stain (ThermoFisher) for 20 min at 4 °C. After washing, single live CD14⁻CD19⁻CD8⁺CD3⁺HLA-A2/YFV-dextramer⁺ cells were sorted directly into 96 well U-bottom plates containing 500 ng/ml HLA-A2/YFV peptide (LLWNGPMAV), 40 U/ml human recombinant IL-2, and 40.000 irradiated (25 Gy) CD3-depleted autologous PBMCs in T-cell media (RPMI1640 with 10% heat-inactivated human AB sera, 1 mM sodium pyruvate, 10 mM Hepes, 50 μM 2-mercaptoethanol, 1 mM L-glutamine, 100 U/ml penicillin and 50 μg/ml streptomycin) and were cultured for 20 days. Every 7 days half of the media was replaced with fresh T-cell media containing 50 U/ml IL-2, 500 ng/ml peptide, and 40,000 irradiated CD3-depleted autologous PBMCs, and the wells were visually inspected for proliferation. Clonal expansions of single HLA-A2/YFV-specific CD8⁺ T-cells clones were confirmed by flow cytometry by using the same staining protocol as described above. Clones with a sufficient number of clonal progeny were subsequently cryopreserved in fetal bovine serum with 10% DMSO and stored in liquid nitrogen until sorting for DNA/RNA sequencing analysis. To isolate single cells from two selected YFV-specific CD8+ T cell clones (A and B), the clones were thawed, washed twice in RPMI1640 supplemented with 10% fetal bovine serum, and stained with the antibodies described above and index-sorted into 96 well PCR plates (Thermo Fisher) or a dMDA cartridge containing lysis buffer as described in the following sections.

### WGA by droplet MDA (dMDA)
The single T-cells were sorted in a FACS instrument equipped with a custom 3D printed adapter holding a dMDA cartridge (cat# CA20100-16, Samplix ApS, Herlev, Denmark) and deposited directly into 2.8 μL lysis buffer (200 mM KOH, 5 mM EDTA (pH 8) and 40 mM 1,4-dithiothreitol) positioned at the dMDA cartridge's Inlet site. The lysis buffer does not contain a fragmenting agent and the protease inhibitor 1,4-dithiothreitol was included to inhibit enzymes that fragment DNA. However, at high temperatures and high pH, DNA may be susceptible to alkaline denaturation. Single cells were lysed for 5 min at room

temperature followed by the addition of 1.4 μL neutralization buffer (400 mM HCl and 600 mM Tris HCl (pH 7.5)) and incubated for 5 min at room temperature. Then, 15.8 μL MDA amplification mixture including polymerase, primers, dNTP, and reaction buffer (Samplix dMDA kit item# RE20300, Samplix ApS, Herlev, Denmark), was added, by injecting it into the dMDA cartridge Inlet site using a wide bore pipette. Finally, 75 μL dMDA oil (Samplix dMDA kit item# RE20300, Samplix ApS, Herlev, Denmark) was added into the inlet well. The dMDA cartridge was moved into the Xdrop™ droplet generator (item# IN00100-SF002 Samplix ApS, Herlev, Denmark) to create single emulsion dMDA droplets. Droplets were collected into low bind 0.2 ml PCR vials from the Collection container of the dMDA cartridge and excess oil was removed from the bottom. The MDA droplets were incubated in a thermal block at 30 °C for 16 h and then heat-inactivated at 65 °C for 10 min and then cooled down to 4 °C. Droplets were broken by adding 20 μL Break solution (Samplix dMDA kit item# RE20300, Samplix ApS, Herlev, Denmark) and the aqueous phase collected containing the amplified DNA. DNA material from Xdrop™ droplet MDA reactions was quantified using Qubit™ Fluorometer (Thermo-Fisher Inc., Waltham, MA, USA) and the DNA integrity was investigated using Fragment Analyzer (Agilent Inc., Santa Clara, CA, USA) according to the manufacturer's instructions.

### WGA by regular MDA

For comparison to the Xdrop™ droplet MDA process, single T-cells were sorted in the FACS and singly deposited directly into 2.8 μL lysis buffer (200 mM KOH, 5 mM EDTA (pH 8) and 40 mM 1.4 DTT) at the bottom of a 0.2 ml PCR vial or 96-well plate. Single cells were lysed, and DNA denatured for 5 min at room temperature followed by the addition of 1.4 μL neutralization buffer (400 mM HCl and 600 mM Tris HCl (pH 7.5)) and incubation for 5 min at room temperature. The MDA reactions were prepared using RepliPHI Phi29 DNA polymerase and Reagent set (Epicentre, Illumina, Madison, WI, USA) according to the manufacturer's instructions. The reactions were carried out at 30 °C for 8–16 h and then heat-inactivated at 65 °C for 10 min.

### Illumina whole-genome sequencing

Illumina libraries were prepared using an automated version of the TruSeq DNA PCR-Free kit. Briefly, DNA was quantified using Qubit HS DNA, and 1 μg of DNA was used as input. The samples were then fragmented using Covaris E220 system, aiming for a fragment size of 350 bp. Fragmented DNA was end-repaired, followed by size selection using Dynabeads MyOne Carboxylic Acid beads. Illumina TruSeq DNA CD Indexes with sample-specific barcode sequences were ligated and the final product was cleaned up using AMPure XP beads. Finished libraries were normalized based on their concentration and sequenced on NovaSeq6000 (NovaSeq Control Software 1.6.0/RTA v3.4.4) with a 2 × 151 setup using 'NovaSeqXp' workflow in 'S4' mode flowcell. Bcl to FastQ conversion was performed using bcl2fastq_v2.20.0.422 from the CASAVA software suite (v1.6).

### Mapping and variant detection in Illumina data

Illumina data was aligned to GRCh38 using BWA mem (v0.7.17-r1188)[63]. The aligned data was sorted using Samtools sort (v1.17)[64] and deduplicated using Picard MarkDuplicates (v2.20.4-SNAPSHOT) (https://broadinstitute.github.io/picard/). Quality control was performed using Picard CollectGCMetrics and Picard WGSMetrics, as well as Samtools flagstats. The analysis was performed on the PCR-free bulk WGS and each of the single-cell samples. The subsequent bam files were searched for SNVs and SVs. The SNV calling was performed using Bcftools call (v1.10+htslib-1.10) and the resulting SNVs were decomposed and normalized using Vt[65]. SNVs overlapping between single-cell and bulk samples were detected by BEDtools intersect (v2.29.2)[66]. SV detection was performed using TIDDIT (v2.11.0)[67] and Manta (v1.6.0)[68]. The TIDDIT calls were filtered based on the Filter column—keeping only

PASS variants. Next, the SV calls were combined using SVDB merge (v2.4.0), combining calls positioned within 200 bp from each other, and sharing an overlap of at least 10% bases. For Illumina single-cell samples, the true positive SVs were defined as the set of events that have an overlap with some SV from the Illumina bulk sample.

### Downsampling and quality control of Illumina data

Downsampling to 100 M read pairs was performed for each Illumina dataset using Samtools view. Thereafter, the coverage was analyzed using TIDDIT cov, computing the coverage in bins sized 5 kbp and 500 kbp across the entire genome. The 500 kbp analysis was visualized using Circos (v0.69.9)[69], displaying coverage levels as a heatmap. The 5 kbp analysis was used to estimate the fraction of reads within high (>200×) coverage regions; the fraction of reads in such regions was computed using Samtools view, searching for reads overlapping high coverage regions as reported by TIDDIT cov. Quality control was performed using MultiQC (v1.12)[70] on all single-cell data (both from Illumina and PacBio) and the results are available as Supplementary Data File S1. The standard deviation of coverage across the genome was estimated using the function bamqc in Qualimap v2.2.1[71] with default parameters.

### PacBio whole-genome sequencing

Five dMDA samples, two from clone A and three from clone B, were chosen for sequencing based on input fragment length and DNA amount. The samples were fragmented to 10 kb using Megaruptor 2 (Diagenode). For each fragmented sample, SMRTbell construction was performed using the Express Template prep kit 2.0. Incomplete SMRTbells were removed using the SMRTbell Enzyme Clean up Kit. SMRTbells were size selected using AMPure beads to remove fragments shorter than 3 kb. The library preparation procedure is described in the protocol "Preparing HiFi Libraries from Low DNA Input Using SMRTbell Express Template Prep Kit 2.0" from PacBio. The SMRTbell library sizes and profiles were evaluated using the Agilent DNA 12000 kit on the Bioanalyzer system. A separate SMRTbell library was prepared from bulk DNA according to Pacbio's Procedure & Checklist−Preparing HiFi SMRTbell® Libraries using the SMRTbell Express Template Prep Kit 2.0. Size selection of the bulk DNA HiFi library was performed using the SageElf system. PacBio sequencing was performed on the Sequel II or Sequel IIe instrument with 30 h movie time. The single-cell libraries were sequenced on one SMRT cell each, while the bulk DNA library was sequenced across three SMRT cells generating 32× coverage of the human genome.

### Mapping and variant detection in PacBio data

PacBio HiFi reads were generated using the circular consensus sequencing (CCS) tool in SMRTLink v10.1. The HiFi reads were aligned to hg38 using version 2.24 of Minimap2[72]. DeepVariant (v1.5.0)[53] was used for SNVs calling and SVs were detected with Sniffles2 (v2.0.7)[54]. TRs were detected through re-alignment of the raw HiFi data using LAST followed by analysis with Tandem Genotypes[55] (v1.1.0). Each analysis was performed individually for the individual T-cells, as well as on the 32× coverage PacBio bulk dataset. SNVs overlapping between single-cell and bulk samples were detected by BEDtools intersect (v2.29.2)[66], while SV overlapping between samples was detected using Sniffles2 multi-sample SV calling. The commands and parameters for PacBio single-cell variant analysis are available in Supplementary Methods.

### Detection of somatic SNVs in PacBio single-cell data

Candidate somatic SNVs were identified through subsequent steps of filtering using BEDtools intersect (v2.29.2)[66]. First, all SNVs occurring in at least two single cells originating from the same T-clone were identified (either A or B). Next, all SNVs that were called in the bulk sample or from a single cell from the other clone were removed. In order to exclude SNVs that may occur due to alignment errors, we further

removed all small insertion/deletion variants as well as SNVs occurring in highly repetitive regions as defined by the GRCh38 RepeatMasker track in the UCSC genome browser. The remaining SNVs were manually inspected in IGV[73] to link each somatic variant to its respective haplotype obtained from the bulk DNA sample.

## Analysis of SV events in PacBio single-cell data

A multi-sample VCF file was generated by Sniffles2 (v2.0.7)[54], where each line corresponds to a unique SV event and where the columns indicate the presence/absence in the different single-cell PacBio samples and the bulk PacBio sample. We further filtered the multi-sample VCF so that only the events annotated as "PRECISE" remained, i.e., SVs with well-defined junctions. This resulting file represents the complete set of SVs across all the samples, and based on this file we could easily apply filters to identify SVs that are shared with the bulk sample. To search for candidate somatic SVs, we filtered out the events occurring in at least two single cells from the same clone (A or B), but not in any single cell from the other clone. The somatic SV candidates were then manually inspected using IGV[73] but no event was found to clearly distinguish between the two T-cell clones.

## Analysis of TRs

To facilitate the analysis of TRs, we focused only on the elements that could be clearly genotyped as either homozygous or heterozygous in the PacBio bulk DNA samples. We, therefore, required that the TRs in the bulk sample should have (i) at least five reads for each of the two alleles for a heterozygous sample, and at least 10 reads for a homozygous sample (ii) no other alleles detected in >2 reads (iii) at least 90% of reads corresponding to the repeat allele(s) (iv) a total coverage of at most 50×. This analysis resulted in 15,098 TRs, which we focused on for the analysis of the PacBio single cells data. In each of the single cells, we analyzed the TR results at the 15,098 sites. Results were reported for all TRs that could be genotyped either as heterozygous or homozygous in the single cells. We further searched for TRs having different lengths in the single cells as compared to bulk DNA but found no such examples.

## Assembly of PacBio single-cell data

The PacBio HiFi reads were first filtered to remove intramolecular chimeras. This filtering was done by aligning each read to its sequence using BLAST[74] and removing all reads that have a secondary blast hit against themselves, at an identity higher than 90%. In this way, reads containing intramolecular chimeras such as inversions and duplications are efficiently removed. The HiFi reads that pass the chimera filtering were then assembled using hifiasm[56] (v.0.7-dirty-r255). The quality of the genome assemblies was assessed using QUAST (v5.2)[75].

## Reporting summary

Further information on research design is available in the Nature Portfolio Reporting Summary linked to this article.

## Data availability

The data is deposited on a secure Swedish server and has been assigned a DOI (10.17044/scilifelab.22730684). Data access requests may be submitted to the Science for Life Laboratory Data Center through the DOI. Source data are provided with this paper.

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

## Acknowledgements

We thank Thorarinn Blondal at Samplix for assistance in setting up the Xdrop system for dMDA experiments. Library preparation and sequencing were performed at the SciLifeLab National Genomics Infrastructure

in Stockholm and Uppsala. Computational analyses were performed on resources provided by SNIC through Uppsala Multidisciplinary Center for Advanced Computational Science (UPPMAX) under the SNIC projects sens2016003 and sens2019020. The project was partially funded by a SciLifeLab Technology Development Project Grant (A.A.), the Swedish Research Council (2019-01976) (J.N.), and the Göran Gustafsson Foundation (J.N.). Parts of the sequencing costs were funded by Samplix.

## Author contributions

J.H. and A.A. devised and supervised the project and were involved in performing the experiments and data analysis. J.E.M., S.H., O.C-L., J.N., C.-J.R., L.F, and J.M. contributed to project planning and wet lab experiments. J.E., C.T.-R., C-S.C., and I.B. contributed to data analysis. J.H. and A.A. wrote the paper.

## Funding

## Competing interests

C.-S.C. is an employee and shareholder of GeneDX, LLC. The other authors declare no competing interests. Parts of the sequencing costs were funded by Samplix.
