## [Peer Review File · Nature Communications]

Long-read whole genome analysis of human single cellsREVIEWER COMMENTS

Reviewer #1 (Remarks to the Author):

In this manuscript, the authors describe a new approach that combines single-cell droplet MDA with long-read sequencing to provide longer regional assemblies and identify structural variants that are missed with short read sequencing. The general concept and approach are of interest to the community, but there are several major concerns.

- 1) As this is a methods paper, there needs to be much more rigor in the evaluation of the data. They should provide general metrics such as coverage breadth and uniformity when comparing short and long reads. The uniformity is typically quantified with the gini index, MAPD, and/or CV of coverage.
- 2) More importantly, the authors should provide sensitivity and precision measurements for each class of variants (SNV, SV, Indel, CNV) and compare those values to the sensitivity and precision of the short read data. The clonally expanded sister cells should enable the accurate measurement of false positive rates as they should be identical. The short read data in figure 3a suggest that the sensitivity of standard and droplet MDA are significantly worse than other published methods (0.5-1.5M SNV's out of 3.5M)
- 3) Much more work needs to be done on characterizing chimeras. It is possible they are just detecting more chimeric artifacts, not true SV. The sister cells should again be used to sort through this.
- 4) The authors should acknowledge the technical challenges in performing droplet MDA. In addition, there is a new method (primary template-directed amplification) that appears to outperform droplet MDA without the need for microfluidics.
- 5) Sequencing should be subsampled to the same number of bp read for all comparisons.

Reviewer #2 (Remarks to the Author):

The cells of organisms are inherently unique, with no two being exactly the same. As such, exploring genetic variation at the single-cell level and elucidating intercellular heterogeneity represent critical undertakings towards understanding the diversity of physiological processes and disease pathogenesis. Even the genome sequence can display subtle variations between normal somatic cells within an individual, necessitating the use of highly sensitive and reliable methods for their detection. Despite utilizing a limited number of cells, the authors demonstrate the feasibility of dMDA combined with long-read whole genome sequencing in identifying prevalent genomic aberrations at the single-cell level, including SNVs, SVs, and TRs. Furthermore, the authors used long-read sequencing data to perform de novo assembly of two single cell genomes and subsequently evaluated the completeness of these assemblies. The above technical demonstration and accompanying data analyses provide useful information for researchers in this field. I have some comments and suggestions below.

1. In the Methods section, the authors should add a few sentences to clarify the role of the lysis buffer (200 mM KOH, 5 mM EDTA (pH 8) and 40 mM 1,4-dithiothreitol) to fragment genomic DNA. How did the authors assess the degree of fragmentation (the number and size distribution of DNA fragments)?
2. It is important that a new method is accessible to the community. Can the authors comment on whether dMDA droplets can be generated using other commercially available droplet generators, such as the QX200 Droplet Generator (a component of Bio-Rad's QX200 Droplet Digital PCR System), which is more common in molecular biology labs?

3. Does the size selection of single-cell PacBio HiFi libraries potentially result in diminished genome coverage and allelic dropout, despite its benefits of elevated CCS read length and yield?
4. The authors do not present a clear analysis to benchmark the performance of each variant type identified within a single cell using PacBio/Illumina data. Precision, Recall, and F1-score are commonly used metrics to assess the accuracy of variant calling results, and the inclusion of these metrics in their analysis would enable a more rigorous evaluation of their findings.
5. The authors should provide a detailed description of their filtering strategies or employ a gold standard workflow (e.g., GATK for Illumina SNV calling) to eliminate false-positive variants in both single cells and bulk samples.
6. I would suggest that the authors utilize the latest version of Minimap2 (v2.24) or Winnowmap (v2.03) to generate alignments for SV calling using long-read sequencing data. These software versions have improved their mapping accuracy within repetitive regions.
7. Additional assessments on the quality of two single cell assemblies need to be conducted, including the computation of the NGAx statistic, the estimation of base-level accuracy through quality value (QV), and an examination of the major histocompatibility complex (MHC) region.
8. With the acquisition of the complete mitochondrial genome assembly, the authors would use NOVOPlasty and Illumina data to perform a more thorough examination and evaluation of heteroplasmy variation between individual cells.
9. In the Methods section, the authors are required to provide detailed information regarding the parameters used for each program or tool.
10. For data availability, if raw sequencing data have been uploaded to the NCBI Sequence Read Archive but not yet made public, the authors should provide a Reviewer link.
11. In Figure S4 and its legend, the number "16,281" should be corrected to "16,218".
12. In Table S1, the amount of data for each library should be given.
13. The size of the reference genome (GRCh38) used for calculating the covered bases (in Table S2 and S4) and the assembly completeness (in Table S11) should be specified.
14. Did the authors intentionally use "SNV" and "SNP" interchangeably in their writing, given the related nature of these concepts within the genomics field?
15. It might be worth considering citing the paper "De novo assembly of human genome at single-cell levels" by Xie et al. (2022), which presents the first single-cell de novo assembly of human genome.

***** We want to thank both reviewers for their constructive critique. Based on their comments we now have a significantly improved manuscript. Please see below for our point-by-point answers to all the comments. In addition to the changes suggested by the reviewers, the abstract has been reduced to 150 words to fit with the manuscript guidelines and minor edits have been made to improve the text. All changes in the manuscript are marked in red. We also included a supplementary figure (Figure S6) that was not requested by the reviewers, which shows the custom 3D printed adapter holding a dMDA cartridge. This supplementary figure was added following a request from a reader of the bioRxiv preprint manuscript version. *****

REVIEWER COMMENTS

Reviewer #1 (Remarks to the Author):

In this manuscript, the authors describe a new approach that combines single-cell droplet MDA with long-read sequencing to provide longer regional assemblies and identify structural variants that are missed with short read sequencing. The general concept and approach are of interest to the community, but there are several major concerns.

1) As this is a methods paper, there needs to be much more rigor in the evaluation of the data. They should provide general metrics such as coverage breadth and uniformity when comparing short and long reads. The uniformity is typically quantified with the gini index, MAPD, and/or CV of coverage.

***** Reply: In the revised manuscript, we have put a lot of effort into improving the rigor of the data analysis and quality assessment since this critique was raised in several of the reviewer comments. Regarding quality evaluation, we have now run the tool MultiQC (<https://multiqc.info>), which compiles the results from several different bioinformatics QC tools into a single HTML report. In the revised manuscript, the HTML file is available as supplementary material so it can be downloaded and browsed interactively by anyone interested to examine the QC statistics in detail. The quality evaluation is mentioned in the results section (lines 136-137) and references to MultiQC and to the supplementary data file has been included in Methods (lines 460-462). *****

2) More importantly, the authors should provide sensitivity and precision measurements for each class of variants (SNV, SV, Indel, CNV) and compare those values to the sensitivity and precision of the short read data. The clonally expanded sister cells should enable the accurate measurement of false positive rates as they should be identical.

***** Reply: We agree that sensitivity and precision are important metrics and we have performed several new analyses to address this point. First, we reanalyzed all samples using the latest software versions (see Reviewer 2, point 6). After having obtained the new results, we performed sensitivity and precision measurements for the different class of variants. For these calculations we decided to use bulk WGS data as the ground truth instead of the clonally**

expanded sister cells as suggested by the reviewer. The reason for this choice is that the sister cell data contain amplification errors and allelic dropout, leading to many missing variants and false positives. The Illumina and PacBio bulk WGS data, on the other hand, should contain virtually all detectable by the respective technology. Even though a few somatic variants exist in the single cells, i.e. not detectable in the bulk WGS data, these will not have a noticeable impact on the overall statistics.

The sensitivity and precision estimates are presented in the results sections for SNVs and SVs. For SNVs, the results are shown in four new panels (Figure 3A-D) as well as in supplementary table S6. For SVs, we constructed a new version of Figure 4 that includes sensitivity/precision estimates for insertions, deletions, duplications and inversions. The SV results are also presented in supplementary tables S10-S14. We believe that the new results are a really important addition to the paper and would like to thank the reviewer for raising this point.

In summary, our results show that PacBio has higher precision than Illumina across all classes of variants. For large insertions and deletions the difference is huge, with PacBio being superior both on precision and sensitivity. However, we also find that virtually all duplications and insertions detected by PacBio are false positives. The only plausible explanation for these false positives is that they occur due to chimeric artifacts from the MDA procedure. We therefore make use of these duplication and inversion calls to gain more information on chimeric artifacts (see also point 3 below). ***

The short read data in figure 3a suggest that the sensitivity of standard and droplet MDA are significantly worse than other published methods (0.5-1.5M SNV's out of 3.5M)

*** Reply: Figure 3a in the old version of the manuscript was not ideal for comparing SNV sensitivity between the different methods. We have now created a new Figure 3 that displays the results in a much clearer way and also gives numbers for the sensitivity. As seen in Figure 3d, Illumina dMDA in fact gives higher sensitivity (0.24) than Illumina MDA (despite lower total data amount for dMDA). This is consistent with our findings on the downsampled data in Figure 2, which shows that dMDA gives less amplification bias and higher overall genome coverage.

This said, there exist other methods that potentially could give even better genome coverage. However, we opted for MDA since this method gives fragments of around 10kb range that are well-suited for PacBio sequencing. To our knowledge, many of the other methods that give higher sensitivity give shorter fragment lengths. For more details on other amplification approaches, please also see our answer to point 4 below. ***

3) Much more work needs to be done on characterizing chimeras. It is possible they are just detecting more chimeric artifacts, not true SV. The sister cells should again be used to sort through this.

*** Reply: To better address the question of chimeras, we performed in-depth analysis of the SVs to determine which ones are true and which ones are false and likely caused by chimeric

molecules. Again, we used the bulk WGS data for this analysis instead of the sister cells (see point 2 above). Our results show that 73% deletions and 66% of insertions are true positives since they are found also in the bulk WGS data. However, we further detect >80,000 additional events across all single cells that are not found in the bulk WGS data. A detailed analysis shows that 81% of these events not detected in bulk DNA are inversions, 9% insertions, and 5% duplications or deletions. We consider these percentages to give a good estimate of the different types of chimeric events in the single cells. In the revised manuscript, we have added a couple of sentences in the SV analysis section in Results (lines 217-220) to present our findings for chimeric SVs. The SV types and lengths for chimeric events are also shown in the new supplementary Figure S5. We believe this new analysis gives a much better overview of the amplification artifacts introduced by the dMDA procedure.***

4) The authors should acknowledge the technical challenges in performing droplet MDA. In addition, there is a new method (primary template-directed amplification) that appears to outperform droplet MDA without the need for microfluidics.

***** Reply: In principle, other chemistries can be used in the droplets. Here, our efforts are focused on long-read single cell sequencing for which long fragment length is crucial.**

To our knowledge, MDA yields the longest fragment lengths among whole genome amplification methods which motivated our decision to utilize this method. While template-directed amplification represents a promising approach for whole genome amplification, it is limited in that it produces reduced fragment sizes corresponding to approximately x bp while the recommended input fragment size for PacBio HiFi library preparation corresponds to ~10 kb. However, we agree with Reviewer #1 that this is an important point as other methods than dMDA may give even better results for future single-cell long-read WGS experiments. On lines 317-322 in the Discussion, we have added a few sentences to mention alternative whole genome amplification methods and also included the following references:

- Fan, X. et al. SMOOTH-seq: single-cell genome sequencing of human cells on a third-generation sequencing platform. *Genome Biol* 22, 195 (2021)
- Biezuner, T. et al. Comparison of seven single cell whole genome amplification commercial kits using targeted sequencing. *Sci Rep* 11, 17171 (2021).
- Borgstrom, E., Paterlini, M., Mold, J.E., Frisen, J. & Lundberg, J. Comparison of whole genome amplification techniques for human single cell exome sequencing. *PLoS One* 12, e0171566 (2017).
- Gonzalez-Pena, V. et al. Accurate genomic variant detection in single cells with primary template-directed amplification. *Proc Natl Acad Sci U S A* 118 (2021). ***

5) Sequencing should be subsampled to the same number of bp read for all comparisons.

***** Reply: In the previous manuscript version we did perform subsampling to compare the two amplification approaches (MDA vs dMDA) in Illumina data. These results, which are presented in the second section of Results and in Figure 2, clearly shows that dMDA gives more uniform coverage and less amplification bias. However, we do not think that subsampling would**

improve the quality of the manuscript when comparing PacBio to Illumina. We obtained only 10 Gb of data for one of the PacBio single-cells and if all datasets were subsampled to this amount we would obtain very poor results, in particular for the Illumina samples. We therefore prefer to present the full dataset. In the revised manuscript we show that even with much more raw data for Illumina, we still obtain way better results with PacBio for SVs and also higher precision for SNVs. In our opinion this is enough to show the benefits of long-read single cell WGS and adding yet another subsampling analysis would not add much value to the manuscript. ***

Reviewer #2 (Remarks to the Author):

The cells of organisms are inherently unique, with no two being exactly the same. As such, exploring genetic variation at the single-cell level and elucidating intercellular heterogeneity represent critical undertakings towards understanding the diversity of physiological processes and disease pathogenesis. Even the genome sequence can display subtle variations between normal somatic cells within an individual, necessitating the use of highly sensitive and reliable methods for their detection. Despite utilizing a limited number of cells, the authors demonstrate the feasibility of dMDA combined with long-read whole genome sequencing in identifying prevalent genomic aberrations at the single-cell level, including SNVs, SVs, and TRs. Furthermore, the authors used long-read sequencing data to perform de novo assembly of two single cell genomes and subsequently evaluated the completeness of these assemblies. The above technical demonstration and accompanying data analyses provide useful information for researchers in this field. I have some comments and suggestions below.

1. In the Methods section, the authors should add a few sentences to clarify the role of the lysis buffer (200 mM KOH, 5 mM EDTA (pH 8) and 40 mM 1,4-dithiothreitol) to fragment genomic DNA.

***** Reply: The role of the lysis buffer is to lyse the cell and nucleus to release genomic material to be distributed in droplets. The lysis buffer does not specifically contain a fragmenting agent, which is why WGA protocols using this lysis buffer may be followed by fragmentation by ultrasonication (Dong et al). 1,4-dithiothreitol is a protease inhibitor and should inhibit any enzymes that fragment DNA. At high temperatures and high pH, DNA may be susceptible to alkaline denaturation due to the abundance of hydroxide ions. We have clarified this in the methods section.**

Dong, X. et al. Accurate identification of single-nucleotide variants in whole-genome-amplified single cells. Nat Methods 14, 491-493 (2017). ***

How did the authors assess the degree of fragmentation (the number and size distribution of DNA fragments)?

***** Reply: The degree of fragmentation following lysis was not assessed prior to amplification. The lysis was performed directly inside of the microfluidic chip and the genomic material was**

subsequently distributed in droplets. The size of the fragments following library preparation was measured and is reported in supplementary figure S1. To our knowledge, there is no instrument available to measure long fragments (up to 10kb or more) in the minute amounts of DNA present in single cells. ***

2. It is important that a new method is accessible to the community. Can the authors comment on whether dMDA droplets can be generated using other commercially available droplet generators, such as the QX200 Droplet Generator (a component of Bio-Rad's QX200 Droplet Digital PCR System), which is more common in molecular biology labs?

***** Reply: This is an interesting and important question. We have discussed it with our contacts at Samplix and they confirm that the methodology could be adapted also to other droplet generator platforms, even though there might be some technical challenges in particular to obtain efficient encapsulation of single-cell DNA fragments into the droplets. However, this is a challenge also for the Xdrop system. Based on this information, we have added the following sentence in the discussion (lines xx-xx): "In this study we used the Xdrop system for dMDA, but the methodology should be possible to adapt also to other droplet generator platforms." *****

3. Does the size selection of single-cell PacBio HiFi libraries potentially result in diminished genome coverage and allelic dropout, despite its benefits of elevated CCS read length and yield?

***** Reply: This is an interesting question. We agree with the reviewer that this is a risk that size selection could lead to loss of some DNA fragments, and to some extent it could explain the allelic dropout in the PacBio samples. In future studies, it would be relevant to evaluate how the size selection affects the genome coverage in order to find an optimal balance between throughput and uniformity. Ideally such experiments should be performed on the Revio system where we would get much higher throughput. Since this study was restricted to Sequel II sequencing, we aimed to maximize the throughput and performed size selection in a similar way as is recommended for low input PacBio samples. *****

4. The authors do not present a clear analysis to benchmark the performance of each variant type identified within a single cell using PacBio/Illumina data. Precision, Recall, and F1-score are commonly used metrics to assess the accuracy of variant calling results, and the inclusion of these metrics in their analysis would enable a more rigorous evaluation of their findings.

***** Reply: The question of precision and recall (sensitivity) for the different classes of variants was raised also by the other reviewer, and it has been addressed in the revised manuscript. For details, please see the reply to Reviewer 1, point 2. *****

5. The authors should provide a detailed description of their filtering strategies or employ a gold standard workflow (e.g., GATK for Illumina SNV calling) to eliminate false-positive variants in both single cells and bulk samples.

***** Reply: We agree that the filtering strategies were not properly described in the previous manuscript version. When analyzing single-cell data from Illumina and PacBio, we use the variants detected in the bulk sample as the ground truth. It is well known that both platforms have high precision for SNVs and in the case of PacBio we also get high-quality SV calls (<https://www.nature.com/articles/s41587-019-0217-9>). To compare SNVs across samples we have used BEDtools intersect. This program can find SNVs that either are shared between VCF files, or alternatively, present in only one sample. Based on this strategy we can identify true positive variants (that are shared with the bulk) as well as candidate somatic SNV events. In the revised manuscript we have implemented a new and easier filtering strategy that is based on multi-sample SV calling by Sniffles2. This is a standard method to find SVs that are overlapping between the samples and therefore a better and more reproducible strategy than our previous approach. From the multi-sample VCF produced by Sniffles2 it is straightforward to identify true SV calls (overlapping with bulk) as well as candidate somatic SVs. We have updated the methods section to give a better description of our variant filtering strategies, and we have also included the commands and parameters for variant analysis in the supplementary material. We believe these changes have improved the quality of the manuscript and would like to thank the reviewer for bringing this point to our attention. *****

6. I would suggest that the authors utilize the latest version of Minimap2 (v2.24) or Winnowmap (v2.03) to generate alignments for SV calling using long-read sequencing data. These software versions have improved their mapping accuracy within repetitive regions.

***** Reply: We would like to thank the reviewer for raising this point. It has taken quite some effort to rerun all the analysis using the latest software versions, but no doubt that it was worth it since it really improved our results and the manuscript as a whole. We now use Minimap v2.24 for alignment, DeepVariant v1.5.0 for SNV calling and Sniffles 2.0.7 for SV calling. The greatest improvement was seen for the SV calls, where we now detect 5,473 high-confidence SVs per single cell instead of 1,278 in the previous version. We did also see some improvements for SNVs. The manuscript and supplementary material has been updated with these new and improved results. *****

7. Additional assessments on the quality of two single cell assemblies need to be conducted, including the computation of the NGAx statistic, the estimation of base-level accuracy through quality value (QV), and an examination of the major histocompatibility complex (MHC) region.

***** Reply: To address this question, we have run the genome assembly quality assessment tool QUAST on our single-cell assemblies. This tool gives detailed information about NGAx statistics, base-level accuracy and many other values. We have included this complete information in a new supplementary Table S18, which is referenced in the Results section (lines 264-276). We also tried to examine the MHC and TCR regions in the de novo assemblies, since they may be of particular interest to study in these human T-cells. However, it is clear that the assemblies are too fragmented and contain too small parts of the regions to obtain any interesting and meaningful results. We have therefore chosen not to include specific results for MHC or TCR in this manuscript. *****

8. With the acquisition of the complete mitochondrial genome assembly, the authors would use NOVOPlasty and Illumina data to perform a more thorough examination and evaluation of heteroplasmy variation between individual cells.

***** Reply: We agree with the reviewer that mitochondrial heteroplasmy is an interesting topic and we already have detected one clear case in the PacBio and Illumina data through regular SNV calling (at position chrM:16,218). Although it in theory would be possible to perform de novo assembly of the Illumina data and possibly identify lower frequency heteroplasmic events, we prefer to focus on the events that occur at higher frequency and that are detectable in the PacBio data. If we were to expand this analysis that would raise new questions about Illumina mtDNA assembly from single cells and detection threshold from single-cell mtDNA, where also factors such as the mitochondrial copy number in each cell, the efficiency of amplification, etc, would need to be taken into account. This would be a very interesting analysis in itself, but we feel that such an analysis falls outside the scope of the current manuscript. *****

9. In the Methods section, the authors are required to provide detailed information regarding the parameters used for each program or tool.

***** Reply: To address this point, we have included a new section in Supplementary Methods "*Commands and parameters for PacBio single-cell variant analysis*", where the exact commands for running the alignment and variant calling tools have been listed. We agree with the reviewer that this is an important addition to the manuscript. *****

10. For data availability, if raw sequencing data have been uploaded to the NCBI Sequence Read Archive but not yet made public, the authors should provide a Reviewer link.

***** Reply: The sequencing data presented in this paper contains sensitive information that cannot be shared openly under the European General Data Protection Regulation (GDPR) and will be shared under controlled access via FEGA Sweden. Information about the datasets included in this study is available at SciLifeLab Data Repository, DOI 10.17044/scilifelab.22730684. The DOI has been included in the manuscript under the section 'Data availability' *****

11. In Figure S4 and its legend, the number "16,281" should be corrected to "16,218".

***** Reply: This has been fixed. *****

12. In Table S1, the amount of data for each library should be given.

***** Reply: This information has been added to Table S1. *****

13. The size of the reference genome (GRCh38) used for calculating the covered bases (in Table S2 and S4) and the assembly completeness (in Table S11) should be specified.

***** Reply: We have updated these tables with the size of the GRCh38 reference genome, which was 3,088,286,401 bp for all these calculations. *****

14. Did the authors intentionally use "SNV" and "SNP" interchangeably in their writing, given the related nature of these concepts within the genomics field?

***** Reply: We thank Reviewer #2 for bringing this point to our attention as this is indeed a typo. All instances of 'SNP' has been replaced by 'SNV' *****

15. It might be worth considering citing the paper "De novo assembly of human genome at single-cell levels" by Xie et al. (2022), which presents the first single-cell de novo assembly of human genome.

***** Reply: This reference has been in the introduction (line 98) *****

REVIEWERS' COMMENTS

Reviewer #2 (Remarks to the Author):

The authors have adequately addressed my comments.

Evaluation of the authors' responses to Reviewer 1's comments: Overall, the authors have made good efforts to incorporate the reviewer's suggestions into the revision. However, there are two points listed below that require further attention.

1. While the authors have provided some QC statistics generated by MultiQC, no measurements of amplification uniformity were made using the Gini Index, MAPD, and/or CV of coverage. Supplementing this part of the analysis would enhance the clarity and strengthen the validity of their findings.
2. Reviewer 1 emphasized benchmarking each class of variants with the clonally expanded sister cells. In the authors' response, it can be inferred that there may not be sufficient sister cells for unamplified bulk sequencing. The authors should explicitly state the reasons behind their decision to employ PBMC bulk WGS data as the ground truth and highlight its limitations in the revised manuscript. Alternatively, if the GM12878 cell line is available to the authors, it could be used to establish false positive rates of the dMDA method.

***** Please see below for our point-by-point answers to the comments. In addition to the changes made in response to Reviewer #2, minor edits have been made to address the editorial comments and to improve the text. All changes in the manuscript are marked in red. *****

REVIEWERS' COMMENTS

Reviewer #2 (Remarks to the Author):

The authors have adequately addressed my comments.

Evaluation of the authors' responses to Reviewer 1's comments: Overall, the authors have made good efforts to incorporate the reviewer's suggestions into the revision. However, there are two points listed below that require further attention.

1. While the authors have provided some QC statistics generated by MultiQC, no measurements of amplification uniformity were made using the Gini Index, MAPD, and/or CV of coverage. Supplementing this part of the analysis would enhance the clarity and strengthen the validity of their findings.

***** Reply: We would like to thank the reviewer for reminding us to include quantifiable results on the uniformity of coverage, something we overlooked in the previous revision round. In the revised manuscript we have assessed the uniformity of coverage using the tool Qualimap. This software calculates the standard deviation (SD) across over the GRCh38 reference. The results are displayed in a new supplementary table (Table S3) and a new supplementary figure (Figure S1). In summary, the Illumina MDA samples has a factor of 2.46 higher coverage SD than the Illumina dMDA samples. This is consistent with our previous observations that dMDA gives increased coverage uniformity as compared to regular MDA. The results are summarized in a new sentence in the Results section (lines 156-158). *****

2. Reviewer 1 emphasized benchmarking each class of variants with the clonally expanded sister cells. In the authors' response, it can be inferred that there may not be sufficient sister cells for unamplified bulk sequencing. The authors should explicitly state the reasons behind their decision to employ PBMC bulk WGS data as the ground truth and highlight its limitations in the revised manuscript. Alternatively, if the GM12878 cell line is available to the authors, it could be used to establish false positive rates of the dMDA method.

***** Reply: We agree with the reviewer that the use of PBMC bulk WGS data as a truth set requires a motivation, including its limitations. To address this point, we have now included the following text in the first section of Results (lines 132-139):**

“Since the bulk datasets represent millions of cells from the same individual, they should contain all germline variation detectable by the respective technology and can therefore be used as germline ‘truth sets’ for both clonal expansions. A limitation with using PBMCs as ground truth is that signals from somatic variants present in founder cells of the individual

clonal expansions will be concealed within the bulk data. While such somatic variants cannot be validated by the bulk sample, they will not have a noticeable impact on the overall performance statistics given the low number of somatic variants as compared to germline variation.” ***